# Cell Phone Radiation Exposure Limits and Engineering Solutions

**DOI:** 10.3390/ijerph20075398

**Published:** 2023-04-04

**Authors:** Paul Héroux, Igor Belyaev, Kent Chamberlin, Suleyman Dasdag, Alvaro Augusto Almeida De Salles, Claudio Enrique Fernandez Rodriguez, Lennart Hardell, Elizabeth Kelley, Kavindra Kumar Kesari, Erica Mallery-Blythe, Ronald L. Melnick, Anthony B. Miller, Joel M. Moskowitz

**Affiliations:** 1Department of Epidemiology, Biostatistics and Occupational Health, Faculty of Medicine, McGill University, Montreal, QC H3A 1G1, Canada; 2Cancer Research Institute, Biomedical Research Center, Slovak Academy of Sciences, 814 38 Bratislava, Slovakia; 3Department of Electrical and Computer Engineering, University of New Hampshire, Durham, NH 03824, USA; 4Biophysics Department, Medical School, Istanbul Medeniyet University, Istanbul 34700, Turkey; 5Graduate Program on Electrical Engineering (PPGEE), Federal University of Rio Grande do Sul (UFRGS), Porto Alegre 90010-150, Brazil; 6Division of Electrical and Electronics Engineering, Federal Institute of Rio Grande do Sul (IFRS), Canoas 92412-240, Brazil; 7Department of Oncology, Orebro University Hospital, 701 85 Orebro, Sweden (Retired);; 8The Environment and Cancer Research Foundation, 702 17 Orebro, Sweden; 9ICBE-EMF and International EMF Scientist Appeal, and Electromagnetic Safety Alliance, Tempe, AZ 85282, USA; 10Department of Applied Physics, School of Science, Aalto University, 02150 Espoo, Finland; 11Physicians’ Health Initiative for Radiation and Environment, East Sussex TN6, UK; 12British Society of Ecological Medicine, London W1W 6DB, UK; 13Oceania Radiofrequency Scientific Advisory Association, Scarborough, QLD 4020, Australia; 14National Toxicology Program (Retired), National Institute of Environmental Health Sciences, Research Triangle Park, Durham, NC 27709, USA; 15Ron Melnick Consulting LLC, North Logan, UT 84341, USA; 16Dalla Lana School of Public Health, University of Toronto, Toronto, ON M5T 3M7, Canada; 17School of Public Health, University of California, Berkeley, CA 94704, USA

**Keywords:** cellular phone, SAR, cancer, electromagnetic hypersensitivity, radiofrequency radiation, antennas

## Abstract

In the 1990s, the Institute of Electrical and Electronics Engineers (IEEE) restricted its risk assessment for human exposure to radiofrequency radiation (RFR) in seven ways: (1) Inappropriate focus on heat, ignoring sub-thermal effects. (2) Reliance on exposure experiments performed over very short times. (3) Overlooking time/amplitude characteristics of RFR signals. (4) Ignoring carcinogenicity, hypersensitivity, and other health conditions connected with RFR. (5) Measuring cellphone Specific Absorption Rates (SAR) at arbitrary distances from the head. (6) Averaging SAR doses at volumetric/mass scales irrelevant to health. (7) Using unrealistic simulations for cell phone SAR estimations. Low-cost software and hardware modifications are proposed here for cellular phone RFR exposure mitigation: (1) inhibiting RFR emissions in contact with the body, (2) use of antenna patterns reducing the Percent of Power absorbed in the Head (PPHead) and body and increasing the Percent of Power Radiated for communications (PPR), and (3) automated protocol-based reductions of the number of RFR emissions, their duration, or integrated dose. These inexpensive measures do not fundamentally alter cell phone functions or communications quality. A health threat is scientifically documented at many levels and acknowledged by industries. Yet mitigation of RFR exposures to users does not appear as a priority with most cell phone manufacturers.

## 1. Introduction

Historically, medical science has rapidly investigated newly available agents such as radiofrequency radiation (RFR) for possible therapeutic applications. As high RFR powers obviously heated tissues, microwaves were evaluated for various therapies such as diathermy and oncological hyperthermia [1]. Quantification of RFR absorption by the body was derived from older thermodynamic and pharmacokinetic concepts.

In the 18th century, Joseph Black noticed that equal masses of different substances needed different amounts of heat to raise their temperature, describing variations in “capacity for heat”. From his observations, the concept of specific heat [2] was derived later.

The absorption rate constant K_a_ is a value used in pharmacokinetics to describe the rate at which a drug enters a system. It is expressed in units of time^−1^.

Early RFR therapies used different terms to specify the amount of RFR energy or power delivered to tissues while relating them to clinical outcomes. However, after 1975 [3,4], the specific absorption rate (SAR), in W/kg, was widely employed not only in medical applications, including ultrasound but also in RFR bioeffect research.

The choice of SAR, rather than electric and magnetic field strengths, as a basic variable for assessments of RFR exposures, was a harbinger of the industry’s focus on RFR heating. Still, the use of SAR was followed by further reductions in the scope of the RFR risk assessment performed by the Institute of Electrical and Electronics Engineers’ C95 Committee [5]. The collection of these restrictions, designated below as “blind spots”, severely restricted the scope of toxicity assessments and impaired protection against RFR-induced health effects. The first four “blind spots” relate to the protection standard, while the last three “blind spots” concern the processes allowing cellular phones into the marketplace.

The rapid deployment of cellular phones in a large proportion of the human population regularly exposed heads and bodies to high levels of completely novel RFR frequencies. The safety limits promoted by the IEEE and the International Commission on Non-Ionizing Radiation Protection (ICNIRP) [6,7,8] have been adopted by many nations despite the fact that these limits were only aimed at protecting workers and the public from acute heating effects of RFR. These limits ignored the non-thermal interactions between RFR fields and the free electric charges present within living tissues.

In addressing the potential health effects of RFR, IEEE focused on its “non-ionizing” property to emphasize its innocuousness. This is highly misleading since living tissues are already and unavoidably ionized [9]. The fact that non-ionizing radiation cannot ionize molecules does not prove that it is harmless. Other mechanisms beyond ionization are at work, producing deleterious health impacts.

The vital process called oxidative phosphorylation is long recognized to depend on sustained mitochondrial electron and proton flows [10]. All life rests on metabolism, which itself depends on the rapid transfer of electrons and protons within or between molecules. It is this electrical traffic, seated in the inner mitochondrial membranes of cells, which is responsible for the generation of adenosine triphosphate (ATP), the energy currency of living cells. Consequently, RFR does not need to reach ionization thresholds to interact with electrical charges and disrupt metabolism. RFR can act on the free electrical charges spontaneously generated by normal physiological processes.

The reduction in oxygen that is part of oxidative phosphorylation inevitably yields reactive oxygen species (ROS) in living cells, due to the leakage of electrons at Complexes I and III, and these leakages are increased by exposure to RFR [11,12].

The wireless industry has often claimed that there is “no mechanism” by which RFR can interact with biological systems at exposures below their thermal recommendations. In fact, the mechanisms are quite clear, involving the perturbation by RFR fields of the movement of charged particles, electrons, and protons that are spontaneously liberated by all living systems. These electric movements, which often occur over very short distances, are ruled not only by Coulomb’s Law but by quantum mechanics and the physics of nonequilibrium and nonlinear systems [13,14]. Living tissues are fundamentally different from metallic electrical conductors in their interactions with RFR. They contain both free electrons and free protons (those of pH), while metals only allow for the movement of electrons. Yet, living systems share an exquisite sensitivity to RFR with metals. Metallic conductors conduct readily because their electrons behave like a degenerate Fermi gas [15]. Biological tissues, like metals, are easily disturbed by vanishingly small (adiabatic) RFR fields [16] by way of the quantum mechanical tunneling of both electrons and protons in respiratory enzymes and in the DNA molecule [17]. This explains why very small RFR fields impact life processes.

These basic mechanisms have a multitude of downstream consequences at an almost unlimited number of points of action within and between important molecules that influence biological events. Life’s processes depend so heavily on the movement of electrical charges that there is no hope of cataloging all these points of action, although science should ultimately identify most of the highly vulnerable sites. It is the multiplicity of these RFR points of biological action that explains why the Oceania Radiofrequency Scientific Advisory Association database on Electro-Magnetic Field (EMF) Bioeffects already lists 400 scientific papers [18] describing altered enzyme activities at power levels that are typical of everyday exposures, much below the safety thresholds promoted by ICNIRP’s industry guidelines.

As examples of important points of action, we will later discuss at the top of Table 1 on page 8 individual steps of electron and proton transfers in chloroplasts and mitochondria in the processes of photosynthesis (plants) and oxidative phosphorylation (animals).

A second category of RFR targets, enzymes, forms the bottom of Table 1. The action of all enzymes, protein interactions, and DNA stability involves hydrogen bonds, which are essentially the delocalization of protons spontaneously migrating between two positions. Six of these hydrogen bond-sensitive processes are listed in Table 1.

There are seven fundamental “Blind Spots” in the IEEE-ICNIRP evaluation of the biological effects of RFR.

## 2. Analysis

### 2.1. Blind Spot #1: RFR Toxicology and Risk Assessment

The principle of Threshold Risk Assessment is to look for transitions as one increases the intensity of exposures on cellular, animal, or population targets. If a sensitive model is available, one may find a no-observable-effect level (NOEL), which represents the exposure preceding the detection of any biological effect of the agent. As one increases exposures, no-observable-*adverse*-effect or lowest-observable-*adverse*-effect levels (NOAEL or LOAEL) can also be detected. Given the limitations of the individual models used to measure these levels, toxicologists prudently consider any biological changes as significant, as their models are restricted in the number of variables monitored, and the time over which observations are performed. In the most favorable situation where the NOEL is determined from experimental observations, it is divided by a Safety Factor to obtain the “Reference Dose” which is thought to be safe for humans. The adopted Safety Factor often compiles factors of 10 to cover uncertainties such as the choice of variables, the use of surrogates in experiments (inter-species), and individual variations (intra-species). If the NOEL is not known, the Safety Factor may be further increased since one then relies on the higher NOAEL or LOAEL levels.

In the late 1960s, when the discussion of RFR safety limits started, there were already many papers documenting non-thermal biological impacts at RFR levels much lower than the current IEEE-ICNIRP safety limits. Contemporary lists of such non-thermal effects are available [19,20]. Acknowledging these reports as genuine “effects” and classifying many of them as “observable” and “adverse” should have lowered the safety levels of RFR by many orders of magnitude. However, IEEE’s C95 committees were reluctant to accept these reports as observable or adverse. This reluctance is well illustrated by the intense attention devoted to heat, with which the engineering community was familiar, and which was falsely promoted as the only genuine or reproducible effect of RFR. This fixation on heat was used to supersede the traditional indicators of toxicity supported by biology and medicine.

To avoid acknowledging them as “observable”, a need for replication was claimed, with strict criteria. In science, studies are very rarely exactly replicated, as replications are generally unpopular with investigators. Not only is funding more difficult to obtain, but it also removes the investigator’s right to originality, discovery, and possibly even patent applications, blighting the image of science moving forward. A second problem is that in bio-medicine, as opposed to physics or engineering, it is almost impossible to reproduce molecular, genetic, cellular, and animal studies exactly. Biological material, in its genetic and microbiome variations, for example, is inherently more difficult to control. A biological model is sensitive to a large variety of variables and can be investigated by a wide diversity of techniques. Placing unrealistic reproducibility requirements on non-thermal RFR biological observations deliberately ignored the practical limits experienced in laboratories when seeking to simulate the physical and biological complexities of the real world [21].

Although there were reliable and repeatedly observed indicators of cell injury signals such as calcium efflux ([22] for a review) and oxidative stress [23,24,25,26] with highly likely connections to human diseases, these studies were not considered by IEEE as indicative of the adverse character of RFR. This opinion runs counter to the advice of the International Agency for Research on Cancer (IARC), for which oxidative stress has been commonly connected to DNA mutations [27].

Perceived problems of validity, reliability, and relevance to human health fueled this skepticism of the industrial community on the heuristic value of physiology and even animal experiments in determining the health risks of RFR. It pushed the center of weight of “acceptable” evidence away from biology towards epidemiology, where causation is even more difficult to establish due to confounding or exposure misclassification. IEEE’s skepticism was then extended to epidemiology when it, too, provided evidence of adverse health effects both at RFR [28] and extra-low frequencies (ELF) [29]. This epidemiological evidence was recently strengthened by two important animal studies [30,31].

The views of the IEEE on biomedical science allowed vast amounts of evidence confirming the health effects of low levels of RF radiation to be labeled by IEEE and ICNIRP as unreliable or not reproduced, and consequently of little value in establishing safety standards.

Despite its rejection of classic risk assessment science, IEEE sought to keep control of the determination of Electromagnetic Radiation (EMR) safety thresholds. IEEE ultimately relied for RFR risk assessment on two behavioral studies conducted by the Naval Aerospace Medical Research Laboratory, one in rats ([32] see the generic setup in Figure 1) and one in monkeys [33] to set a starting point for the determination of a reference dose. IEEE pointed out that in these two experiments, behavioral thresholds for monkeys were similar to those of rats and therefore constituted replication. In view of the species differences, IEEE’s exacting requirements for “replication” were considerably relaxed on this occasion. Other weaknesses of these two de Lorge studies are the very small numbers of animals and exposure sessions (8 rats and 5 monkeys; typically, 3 sessions) as well as their remarkably short durations (40-min in rats and 60-min in monkeys). An even more disturbing question is whether a single acute behavioral response is appropriate for general and long-term toxicity determinations.

Behavioral screening techniques for toxicity determinations were developing at about the same time as de Lorge’s experiments. However, the consensus among scientists in that field at the time was that toxicity thresholds using behavioral methods could only be obtained reliably by using multiple rather than single variables. A historical account of the development of behavioral screening for toxicology [34] states that a battery of tests is needed to find the most sensitive behavioral endpoint for a given agent. The testing arrays required are best described by the Functional Observational Battery (FOB) [35], first published in 1985 by the US Environmental Protection Agency to impose structure in neurotoxicity tests. A FOB should include sensory effects, neuromuscular effects, learning and memory, and the histopathology of the nervous system.

In 1988, a FOB was indeed used to determine the differential toxicity of two pesticides, chlordimeform, and carbaryl [36], but that FOB included no less than 27 variables. Twenty to thirty endpoints are recommended [37], and scientific agencies have come to require or recommend such behavioral evaluations as a part of the standard behavioral toxicity evaluation of chemicals or therapeutic agents.

IEEE’s and ICNIRP’s determination, which underpins current guidance limits, rested on a single outcome, pressing a lever for food, while similar investigations of behavioral toxicity required 20 to 30 different endpoints. It should have been realized that the general toxicity of RFR could not be reliably determined using a test that did not even meet the standards of the day for behavioral toxicity, and certainly not for general toxicity or carcinogenicity. Such a decision could only be justified from an a priori conviction that the only real effects of RFR were thermal.

The rat and monkey tests conducted by de Lorge were 1 hour or less at 0.225, 1.3, and 5.8 GHz and were performed at extremely high doses of RFR. The lowest RFR exposure in any monkey test was 50,000,000 µW/m², and the highest was 1,500,000,000 µW/m². These RFR intensities were so high that they produced minor burns on the faces of three of the five monkeys. These levels are also huge compared to those now considered safe for long-term daytime human exposures by the European Academy for Environmental Medicine (EUROPAEM) working group, 10 to 100 µW/m² [38]. This factor of a million in intensities between de Lorge’s acute data and EUROPAEM recommendations begs the question as to whether the same class of effects was being investigated. Facial burns to radar operators over 1 hour are clearly undesirable in a military context, but are these effects relevant to cancer induction over a lifetime of exposure to cell phone RFR? It is difficult to agree with the notion that the thresholds provided by this monkey data “allow a relatively complete picture of the biological effects of microwaves” as stated in de Lorge’s article. Even the proposed threshold as a function of RFR intensity is dubious since for all frequencies and intensities examined, a delay in response time in the presence of RFR was observed ([33], Figure 3). This indicates that the true NOEL is inevitably smaller than the lowest exposures used within those experiments.

The context of de Lorge’s tests is difficult to reconcile with a general toxicity determination because very high-power densities were used to disturb the animal’s observing responses, and all these responses were exclusively interpreted in a thermal context. Rather, it seems that de Lorge investigated the Maximum Tolerable Doses of RFR heat over 40 min (“40m-MTD”) and related them to a preferred variable, a 1 °C rectal temperature increase in the monkeys, a connection that is itself cast into doubt [39]. In toxicology, MTDs correspond to a dose that does not heavily alter physiology, will not compromise the survival of the animals by causes other than carcinogenicity, or that results in up to a 10% lower body weight after 3 months of exposure [40]. de Lorge’s experiments were investigating tolerance to heat, but could not actually be used to determine general toxicity, NOELs, NOAELs, or LOAELs. The gulf between the RFR safety limits suggested in 1966 by the American National Standards Institute (ANSI), the precursor to IEEE’s C95 committees [41], and those of EUROPAEM 50 years later is explained below.

First, although evidence of non-thermal effects of RFR had been accumulating for decades, a legal challenge to the thermal views of ANSI-1966 and of the commercial practices of the US Federal Communications Commission (FCC) was only successfully mounted in 2020 by multiple plaintiffs including the Environmental Health Trust and the Children’s Health Defense in a US federal appeals court [42].

Second, common exposures in 1966 involved military personnel for field communications and radar operations, and the public, exposed to remote broadcasting towers. Fifty years later, most of the world’s population (4.4 billion in 2017) uses cellular phones routinely placed against the head.

Third, the composition of the committee assessing RFR in 1966 was completely different from later health-oriented groups, reflecting a shift in the population of users. The 1966 representation of wireless “consumers” was mostly from the military (left in Figure 2). The EUROPAEM specialists, 50 years later (right of Figure 2), were devising “guidelines for differential diagnosis and potential treatment of EMF-related human health problems”, essentially involved in the protection of the population now known as the electromagnetically hypersensitive, but earlier designated as victims of “microwave sickness”. Two powerful RFR sources had been progressively added to the environment over 50 years: a source held against the head and the almost ubiquitous network of weaker signals from cell phone base station towers.

At the moment of decision, a 1 °C body temperature rise might have seemed to IEEE as sufficiently equivalent to a slight fever to allay fears of RFR dangers when presented to the public. However, with this focus on heat, IEEE heavily shifted its reference dose towards higher exposures.

IEEE’s 2005 document C95 on RFR safety acknowledges (page 2) that the “rules protect against adverse health effects associated with heating”, and that the safety limits are “designed to protect against adverse health effects resulting from tissue heating, the only established adverse effect of exposure to RF energy at frequencies above 100 kHz”. Further, the document restricts “established health effects” to shocks, “behavioral disruption, heat exhaustion or heat stroke due to excessive whole-body RF exposures”. This document addresses heat as the only “established” effect of RFR and is clearly outdated and wrong based on the many hundreds of scientific studies published on non-thermal effects over the past 25 years [43].

The symbiotic role of IEEE, ICNIRP, and the US FCC in the marketing of wireless devices can be summarized. IEEE, “the world’s largest technical professional organization dedicated to advancing technology” issues health protection limits based on heat. The baton is passed to ICNIRP, which is not a professional organization, but an ad hoc group formed to accommodate the rise in RFR exposures needed by new telecommunications hardware [44,45]. This is carried out by claiming that IEEE recommendations account for all risks beyond heat and that these limits could, in fact, be relaxed. The FCC, a regulator of interstate and international communications, and essentially a spectrum allocation agency with no in-house health expertise, is positioned to set RFR exposure limits in the US. The path of regulation from the 1960s to now is a tunnel mostly insulated from biomedical knowledge that inevitably led to inadequate protection from RFR health risks to humans and the biota.

### 2.2. Blind Spot #2: 40–60 Min Is Not a Lifetime

Notably, de Lorge’s short “performance” experiments could not credibly be extended to cover all possible chronic health effects of RFR, but some data were fortunately available on the long-term thermal adaptation of animals to RFR exposures [5]. This meant that a credible argument to support the high-exposure levels needed for future commercial deployments could be made if the discussion was entirely limited to thermal health effects.

Should de Lorge’s acute exposure experiments (40–60 min) assessing a single outcome, pressing a lever for food, become the basis for general chronic toxicity in humans typically living up to 75 years? The philosophy that allowed this leap of faith by IEEE-ICNIRP is well illustrated by an ICNIRP representative’s comment: “ICNIRP only considers acute effects in its precautionary principle approach. Consideration of long-term effects is not possible” [46]. Essentially, IEEE-ICNIRP believe that long-term effects of RFR are “not corroborated”, not replicated, or not consistent, imposing arbitrary requirements on the reproducibility and reliability of health effects research within the privacy of their committees. Notably, de Lorge’s acute (40–60 min) animal data represent a most extreme case of long-term extrapolation. IEEE used these acute experiments focused exclusively on heat and behavior (Blind Spot #1; [5]) to set safety limits within their C95 committee, while at the same time promoting in other public and government forums that their safety limits encompassed all risks and applied over a human lifetime.

### 2.3. Blind Spot #3: Averaging Human Exposures over 6 or 30 Min

IEEE-ICNIRP’s maximum permissible RFR limits quantify exposures by averaging and compiling 6- or 30-min exposures into a single number. This averaging was justified by the relation between the total energy of the RFR signal and the predictable temperature increase in tissues. However, IEEE-ICNIRP’s averaging procedure erases the rapid variations of the RFR signal over short time intervals, while it is precisely within these short times that biological reactions occur. Transfer of electrons and protons as well as enzymatic reactions occur over time intervals much shorter than 6 or 30 min, as shown in Table 1. This averaging procedure also allows extremely high RFR intensities during very short periods, exceeding the activation levels of specific biological reactions, as discussed below.

**Table 1 ijerph-20-05398-t001:** Transition times in biological reactions cover frequencies from 0.5 Hz to 333 GHz, much faster than IEEE’s RFR integration time.

Biological Events	Execution Time (s)
Individual charge transfer jumps	
Electron and Hole Transfer steps of photosynthesis in Photosystem II [47]	3 × 10^−12^ 2 × 10^−10^ 0.1–2 × 10^−7^ 10^−4^ 0.1–2 × 10^−3^ 10^−2^
Electron and Proton translocations in oxidative phosphorylation Complex IV, cytochrome c oxidase [48]	6 × 10^−5^ 10^−3^ 10^−2^
Enzymatic reaction times from turnover numbers (substrate) [49]	
Carbonic Anhydrase (CO_2_)	2.5 × 10^−6^
Fumarase (Fumarate de- and hydration)	1.25 × 10^−3^
Ribonuclease (RNA degradation)	1.26 × 10^−3^
Tyrosyl-tRNA synthetase (Transfer RNA)	0.13
Pepsin (Protein degradation)	2
Chymotrypsin (Peptide bonds)	7.14
IEEE RFR integration time	360 or 1800 (6 or 30 min)

The second column of Table 1, “Transition times in biological reactions”, lists the execution times of bio-electric charge transfers and enzymatic reactions. If RFR contains frequencies as modulation or carrier components that match the execution times of these processes, interaction is enhanced. Execution times of electron and proton transfers in photosynthesis and oxidative phosphorylation listed at the top of Table 1 correspond to frequencies from 100 Hz to 333 GHz. The execution times of enzyme reactions shown at the bottom of Table 1, protein interactions, as well as the stability of DNA, all involve hydrogen bonds, which are essentially proton exchanges. The enzyme reaction times correspond to frequencies from 0.5 Hz to 0.4 MHz. From Table 1, it can be expected that living systems will display some sensitivity to frequencies ranging from 0.5 Hz to 333 GHz. Such frequencies are widely used in newly deployed power and wireless communications systems. Very brief pulses called “bursts” are almost systematically used to encode information in digital wireless communications, because such strategies benefit signal-to-noise ratio and data capacity. Using an extreme example, a single data “burst” from a Global System for Mobile Communications (GSM) protocol lasts only 0.000577 s. For evidence of harm, IEEE-ICNIRP averages this pulse over 6 or 30 min. This reduces the single burst’s energy by a factor of 623,917 or 3,119,584 times, and its fields by 790 (√623,917) to 1766 (√3,119,584) times. If the energy of an RFR signal is concentrated in a short time but is averaged over a much longer one such as 6 or 30 min, then high instantaneous RFR intensities can disturb biological processes while satisfying IEEE-ICNIRP guidelines.

RFR safety recommendations from health groups such as EUROPAEM [38], the Austrian Medical Association [50], and Baubiologie [51] are based on observation of actual human reactions, and all use RFR exposure estimations based on peak fields rather than averages. This is because peak RFR levels reveal the breadth of biological events that can be activated. As cell phones emit a succession of pulse trains of different durations, the instantaneous fields of such RFR signals allow activation of molecular, electronic, and protonic components within cells that have different thresholds and relaxation times, enriching the non-thermal effects of RFR [52].

Exposure time, although a factor, seems less important than the set of biological reactions that can be activated if one aims to protect a subject from a deleterious environment. Averaging power densities over 6 or 30 min assesses heating, but is insensitive to peak intensities and modulation, even as these peak fields correlate well with neurological impacts, for example.

Biology depends on homeostasis and the large and unpredictable excursions typical of RFR exposures ($\frac{d\left(\mathrm{RFR}\right)}{\mathrm{dt}}$) leave little time for adaptation. Living systems have mostly evolved to deal in their natural environment with the slower rates of change typical of chemical reactions. This aspect of physiological regulation is critical enough in animals that they devote different systems (action potentials, cytokines, and hormones) to manage different time scales.

### 2.4. Blind Spot #4: RFR’s Human Costs

#### 2.4.1. Cancer

Epidemiological studies have reported significant associations between exposure to RFR and increased risks of glioma, acoustic neuroma, and thyroid cancer, among others. Numerous peer-reviewed studies on cell phones indicate that prolonged use leads to glioma as well as acoustic neuroma [53,54,55,56,57,58,59,60,61,62].

That the brains of vulnerable young children are more exposed to cell phone RFR compared to adults [63,64,65,66] and that children will likely use cell phones over their entire lifetime is a major concern. Moreover, the uncertainty in our basic understanding of cancer latencies [67] must be considered when comparing cell phone use with cancer trends.

Meta-analysis of cell phone use and glioma [68] yields an ipsilateral odds ratio (OR) of 2.54, 95% confidence interval, CI = 1.83–3.52, following >1640 h of call time. For meningioma, OR = 1.49, 95 % CI = 1.08–2.06, and for acoustic neuroma, OR = 2.71, 95 % CI = 1.72–4.28, were calculated in the same exposure category. For further details, see Hardell, Carlberg [68]. The attributable fraction (AF) or etiologic fraction is the proportion of cases that can be attributed to a particular exposure. That is the number of cases that would have been prevented if the study variable was not a risk factor. This is calculated as the exposed case fraction multiplied by [(OR−1)/OR]. Based on 247 glioma cases with ipsilateral exposure in Hardell, Carlberg’s Table I [68], AF was calculated as 150 cases, 95 % CI = 112–177. According to Hardell, Carlberg’s Table II [68], 119 meningioma cases reported ipsilateral exposure. AF was calculated as 39 cases, 95 % CI = 9–61. Regarding acoustic neuroma, see Hardell, Carlberg’s Table III [68], 66 ipsilateral exposed cases yielded 42 cases, 95 % CI = 28–51, that could have been prevented.

In older cell phones (1G, 2G; 1980–2000), antennas protruded beyond the top of the units, while the dissimulated antennas of later smartphone models were mostly located at the bottom (3G–4G circa 2001–2022; [69]). This relocated much of the RFR away from the brain, and closer to the thyroid gland. The epidemiological literature linking brain cancer and cell phones peaked around 2011, while fewer recent (~2017) reports link thyroid cancer to cell phones. Keeping in mind that latency should be less in thyroid than in brain cancer [70], the Hardell group found an increased incidence of thyroid cancer in the Nordic countries in the last two decades [69] and confirmed its link to mobile phone use [71]. A case-control study followed by a genetics assessment suggested an increased risk for thyroid cancer associated with long-term use [72,73]. The connection made between cellular phone use and tumors generally [74] only adds to the urgency for exposure mitigation. Although reports of no significant increases in the risk of cancer have been published, it is irresponsible to dismiss the positive case-control studies and endorse negative studies that have limited exposure data, which can result in misclassifications and bias to the null.

EMR is already classified by the International Agency for Research on Cancer in categories 2B (“possibly carcinogenic to humans”) both for ELF [29] and RFR [28], with the RFR classification expected to be reviewed again between 2023 and 2025 [75]. The epidemiological data on cancers are supported by in vivo animal studies, both older [76,77,78] and more recent ones. The National Toxicology Program [30,79,80] and the Ramazzini Institute [31] have both documented that the nervous system is particularly vulnerable to RFR, even within 2 years of experiments on animals.

The adverse effects of EMR at the metabolic and genetic levels have been linked to the induction of reactive oxygen species both in vitro at ELF [81,82] and RFR [22,23,24,25,26]. Increased oxidative stress is a key characteristic of many human carcinogens [27] and can lead to oxidative DNA damage. Beyond cancer and the scope of this article, the effects of RFR are anticipated on human reproduction [83], diabetes, Alzheimer’s, and Parkinson’s diseases, primarily through the established link between RFR and ROS [84,85,86,87,88,89,90].

#### 2.4.2. Electromagnetic Hypersensitivity

The increase in ELF exposures in the last century and of RFR exposures more recently resulted in the emergence of electromagnetic hypersensitivity (EHS) in human populations. Despite exploration and documentation of the EHS syndrome [91], industry and governments have not reacted to curb emissions. Yet, ICNIRP [92] acknowledged that their guidelines may not accommodate the sensitive: “Different groups in a population may have differences in their ability to tolerate a particular non-ionizing radiation (NIR) exposure. For example, children, the elderly, and some chronically ill people might have a lower tolerance for one or more forms of NIR exposure than the rest of the population”.

The victims of artificial EMR (ELF and RF) formed into self-help groups in 18 countries of the developed world, as shown in Table 2, which would have been much longer if not limited to a single entry per country.

The escalating public health importance of this issue was noted by the European Economic and Social Committee. The number of EHS sufferers appears to increase progressively beyond the present 3 to 5% [93]. Current thermal protection has clearly failed this group, which often faces the incomprehension and skepticism of physicians who rely on national recommendations. Not only do EHS sufferers need protection in the form of low-field environments, but preventing an increased incidence of EHS is important in the maintenance of a healthy workforce.

EMR health impacts have spilled beyond humans to agricultural livestock [94] and the environment [95,96,97].

### 2.5. Blind Spot #5: Specific Absorption Rates Determined at Unrealistic Distances

In the earliest telephones, the users listened and spoke into a single hole. Soon, the microphone and speaker became separate units, the first with a fixed attachment, while the speaker, on a wire, was held against the head. Placing the speaker against the head improved sound intensity and excluded the surrounding noise. The handset, combining microphone and speaker, was introduced in 1878. The sound quality was poor in early telephones, but bandwidth gradually improved to the 300–3400 Hz range in current use. In the modern era, with better microphones and speakers, the ergonomics of voice communications improved. Today, speakerphones allow efficient communications with a cellular handset positioned up to about 1 m from the user’s head. The evolution from telephones to cellular phones meant that wireless handsets emitted RFR, first amplitude modulated, but later digitized in the form of pulses. This entirely new environmental agent, potentially exposing large populations to new forms of pulsed radiation, would have needed a serious health impact evaluation by the industry prior to deployment. However, the thermal views of industry meant that recognition of the health consequences of RFR exposure would have to come later, and in great part from academic institutes through epidemiology and animal or laboratory experiments.

The placement of cell phones against the head made the nervous system a primary target, and epidemiological observations were quick to document cancer risks [98]. These risks correlated with mobile phone use, changes in metabolism [99] and the generation of ROS [100]. Even while considering heat only as a risk, anticipated health impacts from RFR prompted the industry to conduct tests designed to limit the heating of the brain by RFR.

In conformity with the view of the industry, a large emphasis was placed on quantifying the heat deposited in the head by a cell phone placed in proximity using experiments on phantoms [101] or by computation [102]. The laboratory experiments (Figure 3) used hollow head shapes filled with a fluid thought to represent the conductivity of human tissues. A probe was used to measure temperature rises due to RFR fields at various locations in the fluid.

Computational strategies based on the Finite-Difference Time-Domain (FDTD) method, by contrast, used computer simulations that were limited not only by resolution but also by the availability of basic data on the anatomy and conductivity/permittivity of real tissues. These early simple models represented the situation depicted at the left of Figure 4, essentially a head together with a phone. Yet it was progressively realized that modeling SAR values with precision at small distances from the head (in the near field) and in proximity to complex layers of biological tissues is difficult and carries an uncertainty of at least 25% [103]. Most notably, SAR decreases by at least 12.5%/mm for very short distances as a cell phone is moved away [104,105,106,107]. Antenna designs and placements within the phone influence SAR, as antennas that radiate power over larger areas [108] or are remotely positioned [109] produce lower SARs.

Even while acknowledging only thermal effects, the industry needed to document the amount of heating produced by RFR as well as how cell phones should be used or positioned, which resulted in the issue of precautionary recommendations to cell phone users. Nevertheless, industry and governments have struggled to control cellular phone emissions, even to meet the permissive thermally based standards of IEEE-ICNIRP [43,103,110]. Table 3 below, compiled from manufacturer’s manuals, lists the minimum recommended distances between cell phones and the body. SAR limits are the same for all but are met at distances that vary from 5 to 25 mm, according to the table. This means that phones are allowed on the market according to a SAR rating based on a protocol chosen by the manufacturer.

In practice, it is difficult for users to maintain the recommended distances, either during a conversation or while simply carrying the device. In other words, most users cannot abide by the recommendations, which are often found deep in the cellphone’s manual. In France, 30 different models of mobile phones with non-compliant SARs have been either withdrawn from the market by the Agence Nationale des Fréquences (ANFR) or have had their specific absorption rate (SAR) updated by software. In addition, the French Agence nationale de sécurité sanitaire de l’alimentation, de l’environnement et du travail (ANSES) recommends that SAR homologation measurements be carried out with the phone in contact (0 mm) with the body, rather than at the distances presented in Table 3. In Canada, Innovation, Science and Economic Development (ISED) confirms that 9 out of 10 cell phones exceed regulatory limits when tested according to “real use” conditions, with the phone in contact with the body (0 mm) [110].

Figure 5 shows that in a scenario similar to Figure 4 and according to the distance between the phone and head, as much as 90% of the power emitted by the antenna is dissipated in the user’s head, as opposed to contributing to communication.

Since the risk of glioma is associated with the cumulative use of cell phones [68,74,111], one would expect SAR reduction to be a priority, meaning that Hardell’s glioma odds ratio of 2.54 [68] should decrease if cell phones were kept at an increased distance from the head/body. This means that the phone-to-head distance is a critical factor in terms of health impacts.

Unfortunately, the question of the distance between cell phones from the head and body is not the only contentious issue in dealing with SAR limits. As discussed below, the volume over which SAR is estimated is also influential.

### 2.6. Blind Spot #6: Specific Absorption Rates Averaged over 1 or 10 g

For estimation of cellular phone SARs, the FCC and IEEE-ICNIRP prescribe averaging to a single value the RFR exposures estimated over 1 or 10 g of tissue, respectively, in the shape of cubes. This implies that over these dimensions, 1 to 2.15 cm (^3^√10), the tissue is uniform in structure and in its sensitivity to RFR, while in fact, it is heterogeneous and anisotropic at the cellular, organelle, molecular, and particle levels. This misrepresentation of living tissues voids any contribution of biology to RFR risk assessment.

A gram of brain tissue may contain 50 million glial cells [112]. Individual exposures to single cells (see Table 4, cell sizes in humans compared to IEEE-ICNIRP averaging), especially stem cells, are significant for cancer risk. The somatic mutation theory of carcinogenesis, the dominant force driving cancer research in the 20th century, proposes that while multiple mutations are usually needed for cancer development [113], even a single DNA mutation can lead to a tumor. Inside cells, small elements such as organelles and enzymes vary in terms of their shape, function, and vulnerability to RFR. The anatomical, electrical, and thermal structures of mitochondria, in particular [114,115], run entirely counter to IEEE-ICNIRP’s philosophy of oversimplification.

Non-thermal effects of RFR are also seen to modify the reaction rates of organic molecules and enzymes in solution [116,117,118,119] or in living cells [120,121]. Electronic and protonic polarizations and their detectable impacts on oxidative phosphorylation charge transfers in mitochondria were already mentioned in the introduction. Inhomogeneities within living tissues from cellular to electronic levels will contribute to characteristics that cannot possibly be addressed by a thermalist view of RFR [49,122] that recognizes living tissues only on a scale of centimeters.

IEEE has historically only considered the properties of different biological tissues when there is a need to relax exposure criteria for specific body parts. For example, equipment already used in industry (dielectric heaters) or by the public (cell phones) needed higher exposures to the extremities (arms) or to the external ear to meet authorization criteria. So, the occasional amendments introduced over time for these tissues increased thermal limits to accommodate the hardware.

FCC-13-39A1 reclassified the external part of the ear (aka pinna) as an extremity [123] despite evidence that DNA damage in the ear canal was connected to cell phone use [124] and despite objections to classifying a body part proximate to the brain as a non-essential “extremity”.

In addition, now, ICNIRP is loosening IEEE recommendations to accommodate 5G, considering a temperature rise of 2 °C to 5 °C as the adverse effect thresholds for the head and limbs, respectively [8]. Those relaxations are unlikely to reflect changing knowledge of human physiology but rather reflect the need for new hardware, the industry’s growing confidence in public demand for wireless communication devices, and its desire to expand the use of cellular phones.

The suggestions of IEEE-ICNIRP in averaging mass/volume are challenged by a series of simulations (Figure 6, [65]) of peak spatial SAR (psSAR) for cell phones as a function of distance to the head. Results show that both the distance of the phone from the body and cube size (or weight, 10 g to 0.01 g) influence psSAR, with smaller cubes producing larger psSARs. Thus, in each 10 g cube, the psSAR somewhere within 0.1 or 0.01 g is much larger than the average. In addition, the results of Figure 6 are far from extending even to cellular sizes (Table 2, Cell sizes in humans compared to IEEE-ICNIRP averaging). Therefore, it is probable that even higher psSAR values would be obtained by reducing cube sizes below 0.01 g. In addition, these estimations, obtained using a four-slab (skin, fat, bone, brain) flat phantom, fall far short of representing the true complexity and anisotropy of cells and mitochondria, both in terms of micro-anatomy and temperature uniformity [114,115].

A given phone’s psSAR rating varies widely, depending on the distance from the head and cube size. The cubes promoted by IEEE-ICNIRP ignore micro-anatomy, which is actually more relevant to the determination of health risks, as it is precisely at the microscopic level that life places the most essential parts of its framework.

The psSAR measurements supporting a cell phone’s access to the market result from a procedure that enables measured emissions to slip under the standard’s safety limits (1.6 W/kg for any 1 g cube for IEEE and FCC or 2 W/kg for any 10 g cube for ICNIRP), rather than from a true scientific determination of exposure or risk.

### 2.7. Blind Spot #7: Enclosing the RFR Source

The SAR estimations of Figure 3, Figure 4, Figure 5 and Figure 6 all feature a cell phone and a head. However, phones are typically held, so why was the hand not considered in simulations? As it turns out, a more realistic image that includes a hand holding a cell phone shows that a substantial proportion of the radiated power dissipates into the body, with a modest remainder actually available for wireless communication. In Figure 7, the Percent-Power-Radiated (PPR in %), which represents the fraction useable for wireless communication as a portion of total emitted power, is simulated using CST Studio with parameters similar to those indicated in Figure 5 and Figure 6. As much as 94% of the power radiated ends up in human tissues in the most common scenario (0 mm).

Since in real-use scenarios, the head and hand together envelop the source, and since the power drawn by conductive/dissipative body tissues is larger than geometrically expected, a cell phone effectively acts as a microwave heater held close to the body. A surprising thermalist opinion which we do not accept even promotes heating people using microwaves within their homes in the winter, as an economical substitute for conventional house heating [125].

The SAR estimation techniques prescribed by IEEE-ICNIRP for cellular phones are defective not only because of distance to head and cube size but also because they have been established with highly unrealistic models.

### 2.8. Blind Spots Summary

Our introduction stated that neither the fact that environmental RFR is non-ionizing nor that its levels are low preclude RFR action on living systems. Non-ionizing radiation does not act by ionizing but acts by direct action on electrons and protons.

IEEE transformed risk assessment into a calorimetric exercise on inert materials while ignoring the processes and structures of life, specifically the motion of electrons and protons, and the properties of protein and enzymes.

Blind Spot #1 illustrates IEEE’s focus on heat, a variable only relevant at extreme RFR intensities, limiting acknowledgment of “real” health effects to the acute, as stated in IEEE [5] and ICNIRP [6] guidelines, and reaffirmed in 2020 [8]. IEEE assumed that a single behavioral change (not pressing a lever for food in rats and monkeys) was the most sensitive and only reproducible effect of RFR exposure, rejecting all other mechanisms.

Blind Spot #2 inappropriately extended the significance of acute RFR experiments to chronic (75 years) situations.

Blind Spot #3 averages RFR exposures over times much longer (6 and 30 min) than the pulses of telecommunications signals, ignoring fast and sensitive biological reactions occurring at peak intensities which are entirely missed by averaging.

Blind Spot #4 denies the real suffering that RFR exposures induced in human populations such as increased tumor rates and electromagnetic hypersensitivity, as well as the environmental effects of EMR.

Blind Spots #5, 6, and 7 illustrate homologation procedures for cellular phone SARs that are not representative of even the thermal risks accepted by IEEE-ICNIRP: distance to the head, 1 or 10 g cubes, and poor representation of actual conditions of use.

The IEEE-ICNIRP has promoted a limited view of the important interactions of EMR with biological systems and health. The seven blind spots reflect a deep misunderstanding of toxicology, biology, and medicine. The deliberate design and promotion of safety limits based on heat is a red herring that avoided the recognition of important RFR toxic effects. Only electric shocks and heating were recognized, thus subverting health effects to the needs of engineering deployments.

The influence of industry extends inside the World Health Organization’s International EMF Project, which anticipates a future where human exposure to EMF “will continue to increase as technology advances” [126]. Since there are no adequate health effects studies on the projected evolution of wireless systems such as 5G [127], and because the effects of 1G to 4G systems have been ignored, one can legitimately ask whether the WHO provides health protection independently from the wireless communications industry [128].

### 2.9. Engineering Solutions

We discuss below three practical approaches to cell phone exposure abatement:Blocking the phone’s RFR emissions, but not its reception, when it is positioned close to the head/body.Modifying the antenna emission pattern (to hemispherical) to radiate away from the head and the body.Limiting call durations according to an estimation of the effective radiated power emitted by the antenna over a specific period.

Those recommendations are germane to those suggested to heavy users [129] such as limiting call durations or keeping the phone away from the head (Figure 4). The most effective exposure reduction measures in hygiene involve control at or by the source, as opposed to reliance on personal habits. Engineering controls, either hard or soft-wired, are effective, while personal habits are less reliable, and need frequent reminders.

Substantial reductions in exposure would be needed to avoid risks of RFR-induced biological effects. For example, glioma risks should be considerably attenuated if the local exposure limit to the head of 1.6 or 2 W/kg was reduced to the whole-body exposure limit of 0.08 W/kg. It is not clear why IEEE-ICNIRP assumes that the glial cells of the brain, oligodendrocytes, astrocytes, ependymal cells, and microglia, are more resistant to RFR exposures than those in the peripheral nervous system or body such as Schwann cells, enteric glial cells, and satellite cells. Why are extremities more resistant to RFR risks than other body parts? A SAR reduction of 20 or 25 times would require much larger distancing, which is achievable by foregoing placing the portable phone against the head, instead keeping it at a distance, and using the speakerphone.

#### 2.9.1. Cell Phone Emission Blocking

Cell phones could be configured to shut off RFR emissions when a proximity sensor detects the presence of the human body. Such body sensors in Android and iPhone devices use a reflected pulsed infrared signal to detect proximity, triggered by the user’s face and switching off the screen, thus preventing any errant soft button presses by the skin, as well as saving battery power. These functions in the Android Application Programming Interface, for example, are under the SensorManager and Sensor classes. With this modification, a user could use the phone normally away from the head, in his hand, or on a table in front. At the cost of a small change in personal habits, this would substantially reduce SAR exposures from cell phone usage.

The network could be designed to alert the user to incoming calls, but the cell phone would be prevented from autonomously sending out data when held against the body, a privacy gain. Essentially, this software adjustment is an automated “Airplane Mode” designed to protect users from radiation [130].

The quality of communication for the hand-held position has increased tremendously since the introduction of speakerphones and microphones integrated into cell phones. The phone against the head position only seems necessary if one must maintain classical phone handling, which is obsolete because of technical improvements. Note that since 2012, all handsets in India must have a hands-free mode [131].

Cell phone blocking is already applied in specific contexts, such as in automotive vehicles, because texting or conversing while driving impairs attention and is a growing cause of car accidents [132]. At least seven “apps” are currently available that use various techniques to prevent texting while driving [133]. The RFR inhibition upon detection of the vehicle’s motion can be provided by the cell phone service provider, by phone-resident software, or even by the car itself, as a virtual barrier around the driver [132]. Cell phone-blocking engineering solutions would save many lives and injuries immediately upon implementation.

#### 2.9.2. Improved Antennas

Apparently, some manufacturers have received patents since the mid-1990s to reduce cell phone radiation to consumers [134]. From 15 different countries, no less than 26 different designs based on Artificial Magnetic Conductors (AMC) and Electromagnetic Band Gap (EBG) structures have been proposed in 25 different publications [135] between 2005 and 2020. Artificial Magnetic Conductors are characterized by negative magnetic permeability, which produces a phase-shifted electromagnetic reflection at a specific frequency. Electromagnetic Band Gap materials use the interference between the direct waves radiated by the antenna and the waves reflected from the material to reduce exposures. The main characteristics of EBG and AMC materials are described by Yang and Rahmat-Samii [136].

Both materials, when underlying antennas, can bring about SAR reductions ranging from 3 dB (6 examples) to 20 dB (14 examples), i.e., energy attenuations of 2- to 100-fold. These materials could be laid into cell phone circuit boards without affecting the quality of communication, and the battery power would be more rationally used in establishing contact with a base station. The general emission pattern would be hemispherical. Superior emission patterns are feasible, as low head SARs have already been implemented by some manufacturers such as Mudita Pure (0.06–0.07 W/kg) [137] and the ZTE Blade V10 [138] at 0.127 W/kg, much lower than the 1.6–2 W/kg exposure limits.

#### 2.9.3. Improved Communication Procedures

Without degrading communications quality, reductions in radiation exposure to users can also be achieved using software capabilities already incorporated in cell phones. Some cell providers currently permit Wi-Fi calling, and a cell phone will switch to that mode of operation when the cell tower signal is unavailable, or when the user has manually selected the airplane mode. A cell phone should automatically use Wi-Fi calling upon the availability of a Wi-Fi connection. The reduction in radiation is realized because communication with the cell tower typically occurs with cell phone transmitter powers between 0.6 and 3.0 Watts, while the cell phone’s Wi-Fi transmitter is generally below 0.1 Watts. This reduction in cell phone transmitted power not only lowers user exposure but also extends battery life.

Another software change uses a cell phone’s existing position-sensing capabilities (accelerometer and GPS) to reduce handshake transmissions with the cell tower. Depending on whether cell phones are used as communication, tracking, surveillance, or policing devices, various levels of handshaking may be desired. Those handshake transmissions allow the cell tower to track cell phone location, and they are currently executed on a regular basis whether the cell phone has moved or not. This increase in RFR traffic [139] is not strictly necessary. Should a cell phone placed on a bedside table emit regular bursts of RFR throughout the night? What is proposed here is that cell phone software be modified so that handshakes with the tower only occur when the cell phone has changed position, reducing radiation exposure and battery drain. Such a change will not prevent the user from receiving a call, as the cell tower will route any incoming calls to the last-known location for the cell phone.

To achieve power savings, most computing devices go into a quiescent mode when a device is not used. Displays, memory units, and processors are placed in sleep mode when their function is not needed, and a similar approach is recommended for cell phones. Unless radiation is necessary, units should drift to radiation levels that are As Low as Reasonably Achievable (ALARA). To realize this goal, airplane mode should be the default for cell phones.

#### 2.9.4. Limiting Call Durations

Beyond the two solutions above, cell phone resident software could limit radiation exposures by controlling cumulative dose, a product of effective radiated power and time. This makes sense since much of the evidence regarding cancer and RFR has used cumulative exposure to establish links. Even if the placement of the phone was allowed against the head, users’ RFR exposure could be automatically controlled by a limit on phone call durations, particularly when base stations are remote. Applications are available to display exposures instantaneously, log them cumulatively, and even control transmissions in high-exposure areas [140].

## 3. Conclusions

Protection measures against wireless RFR exposures need considerable improvement due to the parochial positions adopted in the IEEE-ICNIRP risk assessments. The expanding modern needs for data communications are obviously best served by established optical fiber solutions [141,142] which, in contrast to wireless, offer complete confinement, energy efficiency, and privacy.

Engineering can contemplate many technically practical solutions aimed at reducing cell phone users’ RFR exposures. Software-based solutions controlling RFR emissions, as well as hardware changes to antenna designs, should not be expensive to implement, and would only mildly influence the habits of cell phone users. Although these solutions are available, it seems that in many cases, the industry has either not implemented them, or, in some cases, has even fought exposure abatements by preventing public education about RFR exposures [143,144].

The charters of professional organizations in the world, including engineering, usually state that they place human safety above all other considerations.

We firmly believe that RFR exposures to living tissues should be avoided when possible and that RFR power absorbed by the user’s body is wasted and harmful to health. In all likelihood, our recommendations for cell phone alterations would improve the lifespan of both humans and batteries.

## Figures and Tables

**Figure 1 ijerph-20-05398-f001:**
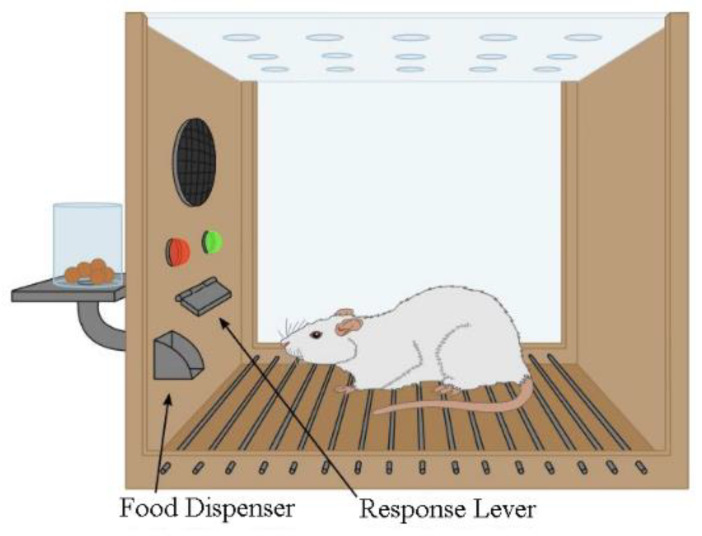
A version of a Skinner box for operant conditioning. The variable discussed is the rate at which a lever is pressed for food by the hungry animal when exposed to RFR.

**Figure 2 ijerph-20-05398-f002:**
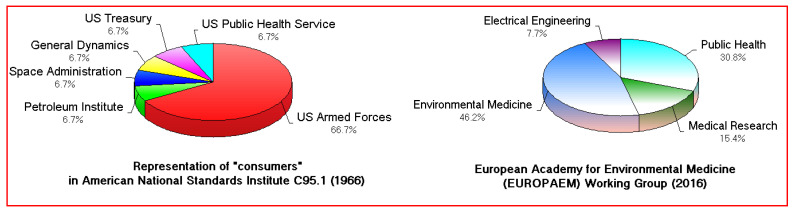
Task forces assessing biological impacts of RFR in 1966 (IEEE) at left, and EUROPAEM, 50 years later, at right.

**Figure 3 ijerph-20-05398-f003:**
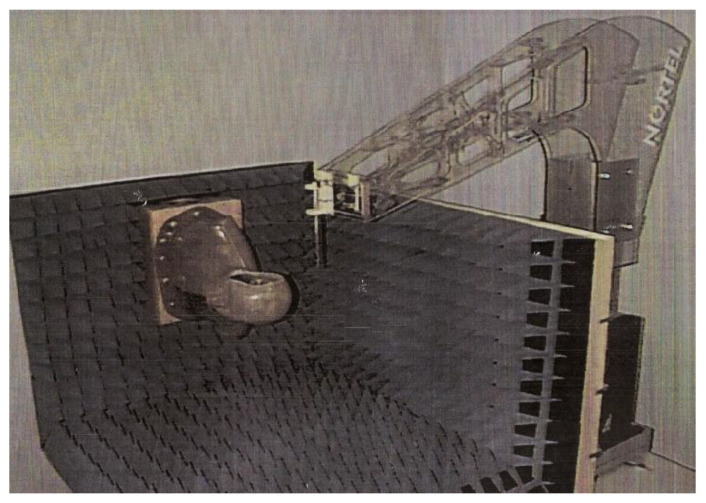
Experimental measurements of SAR used simple head shapes filled with homogeneous fluid.

**Figure 4 ijerph-20-05398-f004:**
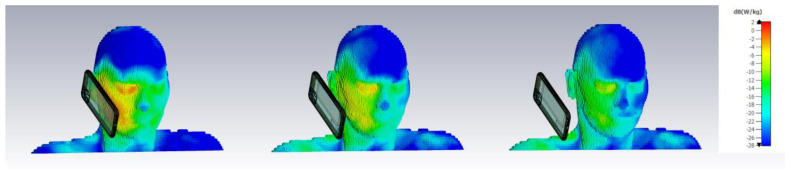
Placing a cell phone antenna at 0, 2, and 6 cm from the head reduces the Percentage of Power absorbed by the head (PPHead, %). 0 dB is 1 W/kg Specific Absorption Rate. Figure 4, Figure 5, Figure 6 and Figure 7 are simulations from Bio extension 4.3 of CST Studio Suite.

**Figure 5 ijerph-20-05398-f005:**
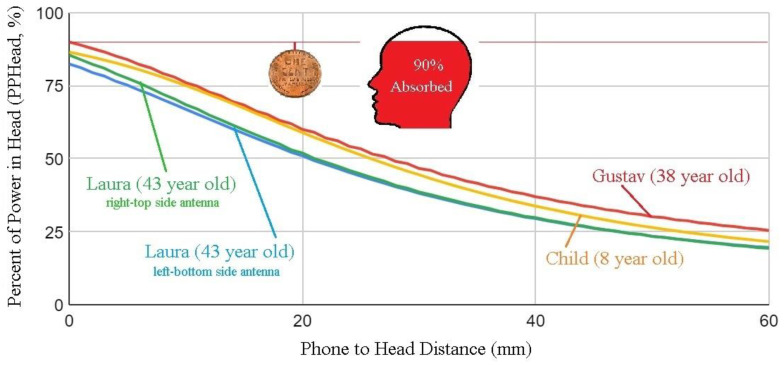
Percent of Power absorbed by the user’s Head (PPHead, %) vs distance of phone to head for models “Laura”, “Child” and “Gustave” using the Dassault Systèmes Simulia Academic CST Studio Suite Research Base Pack 2022 with CST Studio Suite Research Bio Model and CST Studio Suite Research Acceleration Token, purchased from Smarttech, Inc., 2503 Kilgore Street, Orlando, FL 32803 (https://www.3ds.com/products-services/simulia/products/cst-studio-suite/, accessed on 19 January 2023). The cell phone antenna, from the software Antenna Magus, included in the CST Studio package, is a Planar Inverted-F-type Antenna, as is the case for Figure 6 and Figure 7. Placement is in different corners of the cell phone: left-bottom and right-top as indicated for Laura; left-bottom for Gustav and Child. The edges of the antenna and of the cell phone are parallel and the gaps between the antenna and external edges of the cell phone are 2 mm in the 3 axes. The cell phone model is a simple plastic box 147.2 mm × 74.56 mm × 10.48 mm. Only the antenna is powered, no electronic circuit is included in the model. Frequency is 900 MHz. Size of a US penny is shown for distance reference (19 mm).

**Figure 6 ijerph-20-05398-f006:**
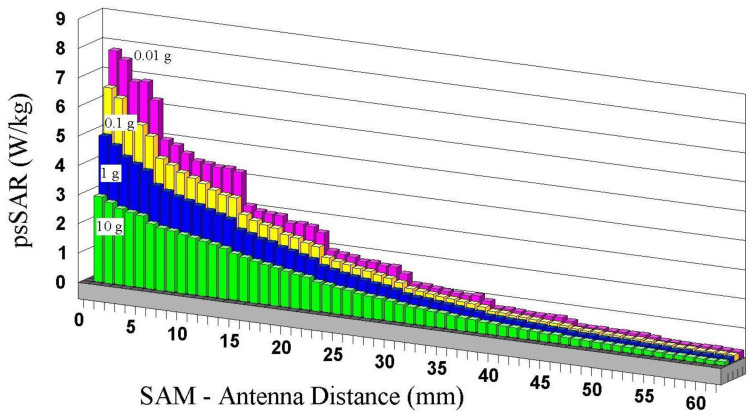
Peak Spatial Specific Absorption Rates (psSAR, W/kg) as a function of distance for four averaging masses. Frequency is 900 MHz and power 250 mW. Tissue and shell parameters, the shape of the Simulated Anthropomorphic Model (SAM) model (IEEE 1528), and voxels from a meshing tool with at least 10 voxels per wavelength are provided by CST Studio. Planar Inverted-F Antenna from Antenna Magus. Cell phone model is an empty plastic box with an antenna 2 mm away from the top edge. The irregularities in the curves stem from a mismatch between head shape and digitized voxels. ICNIRP uses 10 g, while IEEE-FCC uses 1 g. After [65].

**Figure 7 ijerph-20-05398-f007:**
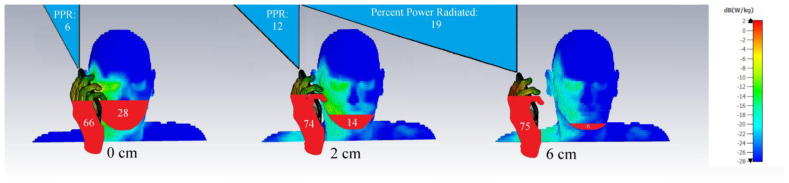
When including the hand holding the phone in psSAR simulations, the 0 cm position (touch) results in 66% of the power being absorbed by the hand (PPHand), 28% by the head (PPHead) for a total absorption of 94% in the body, PPB), and 6% (PPR) radiated for communication. At 2 cm, PPHand is 74%, PPHead is 14%, and PPR is 12%. At 6 cm, PPHand is 75%, PPHead is 6%, and PPR is 19%. The frequency of operation is 900 MHz, and the power delivered to a planar Inverted-F Antenna is 250 mW.

**Table 2 ijerph-20-05398-t002:** Electrosensitive Groups spontaneously formed throughout the world. All sites were accessible on 19 January 2023.

Australia	https://anres.org/information-for-individuals/
Belgium	https://www.bbemg.uliege.be/electrosensitivity-ehs/
Brazil	http://www.slowphone.org/
Canada	http://www.weepinitiative.org/index.html
Denmark	https://EHSF.dk
Finland	https://sahkoherkkyyssaatio.fi/in-english/
France	https://www.electrosensible.org/b2/
Germany	https://www.buergerwelle.de/
Ireland	https://es-ireland.com/about-2/
Italy	https://www.elettrosensibili.it/
Netherlands	http://www.stichtingehs.nl/
Norway	https://www.felo.no/nyheter/
Portugal	http://antenasaquinao.blogspot.com/
Spain	https://www.avaate.org/
Sweden	https://eloverkanslig.org/
Switzerland	https://www.buergerwelle-schweiz.org/
United Kingdom	https://www.es-uk.info/
USA	https://www.meetup.com/emf-awareness-stop-the-stress/

**Table 3 ijerph-20-05398-t003:** Cell Phone Manufacturers’ Recommended Body Separation Distances.

Brand	Cell Phone Model	Separation Distance from the Body in mm
Apple iPhone	11, 12, 13, SE, X, XR, XS	5
Huawei	Y6P	15
LG	G2	10
LG	G3	15
Motorola	razr	25
Nokia	8110 4G	15
Samsung	Galaxy S5	15
Samsung	Galaxy Note 3	10
Samsung	Galaxy Z Fold3 5G	15
Xiaomi	12X	5

**Table 4 ijerph-20-05398-t004:** Cell sizes in humans compared to IEEE-ICNIRP averaging.

Cell Type	Average Volume (µm^3^)	Average Weight (ng)
Sperm cell	30	0.030
Red blood cell	100	0.1
Lymphocyte	130	0.13
Neutrophil	300	0.3
Beta cell	1000	1
Enterocyte	1400	1.4
Fibroblast	2000	2
HeLa, cervix	3000	3
Hair cell (ear)	4000	4
Osteoblast	4000	4
Alveolar Macrophage	5000	5
Cardiomyocyte	15,000	15
Megakaryocyte	30,000	30
Fat cell	600,000 (0.0006 µL)	600
Oocyte	4,000,000 (0.004 µL)	4000
IEEE-ICNIRP 1 or 10 g	1,000,000,000,000 to 10,000,000,000,000	1 or 10

From: http://book.bionumbers.org/how-big-is-a-human-cell/, accessed on 19 January 2023.

## Data Availability

Data is contained within the article.

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
