# Peer review of "Cell Phone Radiation Exposure Limits and Engineering Solutions"

_ijerph, 2023, doi:10.3390/ijerph20075398_

Round 1

Reviewer 1 Report

My expertise is in engineering and my comments will focus only science and engineering aspects of your paper.

The paper should use engineering units throughout, though as authors you may think that adding many zeros to each of the numbers emphasizes more the differences you hope to highlight having to count all the zeros to ascertain what the number really is detracts from the paper intelligibility, is this science? If so use scientific or engineering notation.

Line 229, state the frequency of the tests as this is as important as the power density or 50 W/m2 or 1.5 kW/m2 in determining the exposure of the animal, furthermore these tests were performed by the military with different perspectives. The reference for the paper is not given in the text.

Lines 353-357, this example is factitious whether it is applied to mobile phone or base station (or other type of transmitter), both have limited maximum output powers and could never generate peak powers 100s of thousands or millions of times larger than the average. Peaks can be higher obviously, but limited by transmitter peak powers.

Line 502, remove the information about the computer, it adds nothing.

Line 511, it should be noted that only phones obtaining certification prior to 2016 were allowed to have recommended spacings to the body of up to 25mm and after that the FCC and EU directive RED require the maximum spacing to be 5mm (other regions of the world may not require this)

Table 3, contains phones as old as 2005, I suggest that the oldest phones be removed from the table as they are very unlikely to still be in use.

Line 515, it should be made clear that all phones are tested on the head  touching both ear and cheek and the spacings in table 3 only apply to the phone on the rest of the body, so comment about having to maintain the recommended distances during a conversation is misleading unless using a hands free kit and the phone is on the body.

Figure 5 is at best misleading, and without reference or technical details should be removed. Though it is undoubtedly possible to design an antenna which can achieve this claim, no phone could enter the market for two reasons, it would not meet SAR safety standards and would not be acceptable to the operators as it could not meet the total radiated power TRP requirements for efficient communication with the base station. The results give no indication of the antenna topology (including the rest of the phone), frequency, head model etc.

Line 574, the ear canal is not part of the pinna yet is associated in your text.

Line 578, It would be useful for the reader if the authors detail the safety margins built into the standards for the general public with respect to these values for temperature rise that are given here. Furthermore, it would be great to know in which countries the new limits have been introduced into law, if any.

Figure 6, the results are not from [64] but are of unknown origin, there are no details of the phone model, power, frequency, voxel size, simulation tool, tissue parameters, shell parameters etc. This is unacceptable in a scientific paper.

Line 614, there are multiple papers by these authors which one do you refer to?

Line 684-694. There is a misunderstanding exhibited here. Local SAR limits are the same for the whole body be it head or torso, the whole body average is just that, the power absorbed in the whole body divided by the whole body weight, it is not a reduced local limit as you suggest in line 689.

Section 2.9 should be split into two parts, engineering and phone configuration changes and suggested user behavioral changes

Section 2.9.1 it should be noted that many newer phones already include proximity sensors to regulate the average output power when close to the body to ensure compliance, however, they do not shut off completely as the authors suggest. Line 705, nor would it be possible for the network to know where the phone was if it never communicated with the network when crossing cell boundaries because it was close to the body and “shut off”, so would not be able to alert about incoming calls. The data from apps can be controlled in many cases if the user takes time to adjust settings. Other suggestions are not engineering solutions but changes in behavior, such as using hand free devices or speaker phones.

Section 2.9.2. Antenna design can clearly contribute, however, many papers fail to consider the rest of the mobile phone sub systems that need integration into the small package, or the number of antennas required to cover all the bands and as such academic papers claiming big reductions should be viewed with a certain skepticism. Users actively seeking out SAR data when making a choice of phone should also go into behavioral changes.

Section 2.9.4. is not an engineering solution (apart from possibly social engineering).

Line 784, how does traffic over an optical fiber from a fixed IP offer greater privacy? Maybe remove privacy from this statement.

Reviewer 2 Report

Manuscript ID ijerph-2203489

entitled "Cell Phone Radiation Exposure Limits and Engineering Solutions"

Comments to the Author

The presented review draws attention to the need to significantly improve the protection mechanisms against wireless RFR exposures and supports it with the comprehensive literature survey. The author finds the IEEE's focus on the "non-ionizing" aspect of RFR rather misleading as living tissues are already and inevitably ionized. The authors also reject the wireless industry's assertion that RFR cannot interact with biological systems at dosages below recommended levels because there is "no mechanism" by which this might happen. The author list important action via Table 1 which is very useful. Especially, the author's analysis of 7 important blind points by supporting them with the comprehensive literature survey will shed light on the studies to be done in this field. The authors suggest some solutions such as engineering solutions, software-based solutions. It is emphasized that although these solutions are accessible, it appears that industry has frequently either failed to adopt them or, in some circumstances, actively opposed exposure reduction by obstructing public awareness of RFR exposures. The authors believe that RFR power absorbed by the user's body is wasted and damaging to health, and that RFR exposures to live tissues should be avoided whenever possible in addition to cell phone alterations that they suggest would increase battery life as well as human longevity. When the presented review is examined as a whole, it can be said that it is extremely comprehensive. The topic is very original. It is well-organized. All the sections of the presented review are well-developed. The manuscript is very well and interestingly written, using good English. The literature survey is comprehensive, the references are appropriate. The conclusion is clear and consistent with the arguments presented. Graphical presentation of images meets the criteria of high resolution and readability, legends are exhaustively detailed (which I appreciate). I only suggest to the author to add more clearer image for Figure 3. The presented review is suitable for publication as it is presented.

Reviewer 3 Report

Despite the high number of references provided, this paper presents many statements that are strongly biased and misleading because:

1. they are not justified by providing evidences from proper peer-reviewed scientific literature but are based on personal judgements of the authors or are justified by not scientifically sound references (e.g., articles published in newspapers)

2. if justified by literature, only the literature that supports the authors' arguments is reported without reporting all the remaining literature that sustains a different point of view

3. in a number of cases, the evidences reported in the cited literature are misinterpreted.

Below a few example of such biased statements and misinterpreted literature:

- line  45: justify this sentence with scientific evidences

- l. 297-299: this is a strong and biased statement that implies a misbehaving of IEEE/ICNIRP. Must be justified with objectives evidences.

- l. 355. This statement is not true. IEEE-ICNIRP recommendations are set not only to limit the exposure in 6-30 minutes but also for shorter durations <6min. They consider also the effect of pulsed exposure and the effect of the duration of exposure for exposures <6min.

- l. 487-489; l.633. This statement is not true. Currently available human models are discretized at a high resolution and are electromagnetically characterized for tiny tissues and organs with proved and experimentally validated data. See e.g. human models from the Virtual Population (ViP).

- l.528. How this PPHead value was calculated? Details on the methods and procedure used are missing.

- l.532. which are the model used? Details on their resolution and characteristics are missing.

- l.539-541. This sentence is misleading. Add details and references to current literature to justify this issue related to the volume of calculation.

- l.548. This sentence on misinterpretation of IEEE/ICNIRP of living tissues exposure is biased and not justified by objective and scientifically sound evidences. Also, it disregards the scientific justification proved by IEEE/ICNIRP on the choice of the averaging volume.

- l.566-571; l.580-581; l.603-606. These statements are not true and imply that the work of international committees like IEEE and ICNIRP are not valid and scientifically proved by independent scientists. The authors should thoroughly justify this sentence.

- l.589-590; l.600-601. These sentences are misleading and biased. There is not misbehavior in using 1g averaging tissue in current IEEE and ICNIRP recommendations. Please see the scientific evidences provided by IEEE/ICNIRP to justify this choice.

- l.614-616. Explain in detail how these simulations were done.

- Sect. 2.8. Mostly based on biased argumentations (see comments above)  

- l.685-686. The statement must by justified by strong scientific evidence.

Round 2

Reviewer 3 Report

The authors provided adequate details on the procedure they used to derive the data shown in Fig. 5, as requested.

A part from that, and despite their replies, the critical points evidenced in my previous review still remain. The point is again the same: presence of misleading statements not justified by proper scientific eviidence; presence of personal believes not justified by proper scientifice evidence and adequate discussion among the scientific community.
